# Arbovirus Epidemiology: The Mystery of Unnoticed Epidemics in Ghana, West Africa

**DOI:** 10.3390/microorganisms10101914

**Published:** 2022-09-27

**Authors:** Eric Agboli, Alexandru Tomazatos, Oumou Maiga-Ascofaré, Jürgen May, Renke Lühken, Jonas Schmidt-Chanasit, Hanna Jöst

**Affiliations:** 1Bernhard Nocht Institute for Tropical Medicine, WHO Collaborating Centre for Arbovirus and Haemorrhagic Fever Reference and Research, 20359 Hamburg, Germany; 2University of Health and Allied Sciences, Ho PMB 31, Ghana; 3Kumasi Centre for Collaborative Research in Tropical Medicine, Kwame Nkrumah University of Science and Technology, PMB, Kumasi 039-5028, Ghana; 4Faculty of Mathematics, Informatics and Natural Sciences, Universität Hamburg, 20359 Hamburg, Germany

**Keywords:** arbovirus, vector, outbreak, epidemic, mystery, Ghana

## Abstract

It is evident that all the countries surrounding Ghana have experienced epidemics of key arboviruses of medical importance, such as the recent dengue fever epidemic in Burkina Faso. Therefore, Ghana is considered a ripe zone for epidemics of arboviruses, mainly dengue. Surprisingly, Ghana never experienced the propounded deadly dengue epidemic. Indeed, it is mysterious because the mosquito vectors capable of transmitting the dengue virus, such as *Aedes aegypti*, were identified in Ghana through entomological investigations. Additionally, cases may be missed, as the diagnostic and surveillance capacities of the country are weak. Therefore, we review the arbovirus situation and outline probable reasons for the epidemic mystery in the country. Most of the recorded cases of arbovirus infections were usually investigated via serology by detecting IgM and IgG immunoglobulins in clinical samples, which is indicative of prior exposure but not an active case. This led to the identification of yellow fever virus and dengue virus as the main circulating arboviruses among the Ghanaian population. However, major yellow fever epidemics were reported for over a decade. It is important to note that the reviewed arboviruses were not frequently detected in the vectors. The data highlight the necessity of strengthening the diagnostics and the need for continuous arbovirus and vector surveillance to provide an early warning system for future arbovirus epidemics.

## 1. Introduction

Arthropod-borne viruses (arboviruses) are a category of viruses comprising various taxonomic groups that are transmitted to vertebrate hosts via blood feeding by arthropods, such as mosquitoes, sand flies, ticks, and biting midges [1,2]. Arboviruses are an increasing public, veterinary, and global health concern, and West Africa is predicted to be severely affected in the near future by arboviral diseases [3]. Vector-borne diseases together account for approximately 17% of the estimated global burden of infectious diseases and cause more than 700,000 deaths yearly. Tropical and subtropical areas carry the highest burden of vector-borne diseases. More than 80% of the global population lives in areas at risk from at least one major vector-borne disease [4]. The medically important arboviruses that caused outbreaks and also case fatalities include dengue virus (DENV), yellow fever virus (YFV), chikungunya virus (CHIKV), Zika virus (ZIKV), Rift Valley fever virus (RVFV), and West Nile virus (WNV). Several outbreaks of DENV and YFV have been reported in West Africa, especially in Burkina Faso and Côte d’Ivoire with Ghana reporting only YFV outbreaks [5]. Generally, the epidemiological situation regarding arbovirus transmission in Africa (prevalence and intensity of outbreaks) is different in Asia and the Americas, where many arboviral infections are reported annually compared to Africa [6].

Arboviruses require hematophagous arthropod vectors (mosquitoes, ticks, sand flies, and midges) to transmit viruses between vertebrate hosts [2]. Most of the mosquito-borne viruses affecting humans belong to three families, the *Flaviviridae* (genus *Flavivirus*), *Togaviridae* (genus *Alphavirus*), and the *Peribunyaviridae* (genera *Orthobunyavirus* and *Phlebovirus*) [5,7,8]. 

Generally, mosquitoes are considered the most relevant vectors of arboviruses. Different mosquito taxa such as *Aedes* spp., *Culex* spp., *Mansonia* spp., and *Anopheles* spp. are known to be responsible for transmitting and/or maintaining arboviruses in nature [5]. However, *Aedes aegypti* is the most important vector known to transmit arboviruses of major public health concern. *Ae. aegypti* is a competent vector for ZIKV in Africa, the Americas, Asia, and the Pacific [9]. Infected *Aedes* mosquitoes can also lead to the spread of chikungunya and yellow fever. *Aedes albopictus* is competent for several arboviruses such as DENV, WNV, Japanese encephalitis virus (JEV), CHIKV, Eastern equine encephalitis virus (EEEV), Potosi virus, Cache Valley virus (CVV), Tensaw virus, Keystone virus, La Crosse virus (LCV), Jamestown Canyon virus (JCV), and Usutu virus (USUV) [5,10,11]. Regarding *Cx. quinquefasciatus* mosquitoes, the following viruses are related to the vector: CHIKV, JEV, St. Louis encephalitis virus (SLEV), Western equine encephalitis virus (WEEV), EEEV, WNV, and USUV. To date, WNV is the only virus known to be associated with *Cx. quinquefasciatus* in West Africa [5]. Bunyamwera virus was reported in *Mansonia africana* mosquitoes [12]. *Anopheles* mosquitoes were also reported to harbor RVFV [13] and WNV [14]. Mosquitoes and their breeding sites pose a significant risk for arbovirus infection. Prevention and control rely on reducing the mosquito population through source reduction (removal and modification of breeding sites) and reducing contact between mosquitoes and people.

Arboviral diseases were considered to be only minor contributors to global mortality and disability for decades. Hence, little precedence was given to arbovirus research investment and related public health infrastructure. Reports have shown the unprecedented emergence of epidemic arboviral diseases (particularly dengue, chikungunya, yellow fever, and Zika) in the past few years. These epidemics were due to the triad of the modern world: urbanization, globalization, and international mobility [15]. The outbreak of arboviruses such as dengue was reported in Burkina Faso, Côte d’Ivoire, and Nigeria [16]. Additionally, there were reports in Togo regarding the presence of dengue [17]. These are neighboring countries of Ghana in the western part of Africa. Therefore, Ghana is considered a suitable zone for future outbreaks of arboviruses, mainly dengue. However, this was not the case for Ghana, although a deadly dengue outbreak was predicted [18,19]. Surprisingly, the proximity and high density of mosquito vectors such as *Ae. aegypti*, which are capable of transmitting dengue in Ghana, could not lead to outbreaks. This article reviews arboviruses reported in the literature and the probable reasons for the absence of outbreak reports in Ghana. 

## 2. Materials and Methods

Available peer-reviewed literature about arboviruses investigated or reported in Ghana were systematically compiled from the PubMed (https://pubmed.ncbi.nlm.nih.gov/; accessed on 30 June 2022) and Google Scholar (https://scholar.google.com/; accessed on 30 June 2022) databases. Search terminologies such as ‘arboviruses’, ‘Ghana arboviruses’, ‘Ghana mosquito-borne viruses’, and ‘Ghana mosquito-associated viruses’ were used in the searches. Other sources of information included were previous reviews about arboviruses in the West African region [5,20]. Additionally, case reports related to arboviruses in Ghana were also included in this review. The search revealed publications as early as the 1950s and until June 2022. 

## 3. Results

During the search, a total of 60 records were reviewed and summarized in Table 1. Year of publication was plotted in Figure 1. Additionally, Figure 2 illustrates the map of Ghana with agroecological zones showing the distribution of the reported or detected arboviruses. The evidence for these reports was obtained through seroprevalence studies via IgG/IgM or the detection of arboviral antigens or RNA via RT-PCR. In reports where samples from various regions were pooled together for investigation, and arboviral antibodies or RNA are detected, the distribution goes to all the related regions unless specified. It is important to note that the positions of the viruses, as indicated on the map, are not the exact spot where the respective viruses were reported/detected. Table 1 provides detailed information on the arboviruses investigated or reported in humans and vectors in Ghana.

## 4. Discussion

### 4.1. Medically Important Arboviruses in Ghana

The two main arboviruses of medical importance are YFV and DENV. These viruses seem to be distributed throughout the country. 

#### 4.1.1. Yellow Fever Virus

This virus belongs to the *Flaviviridae* family, and infections occur in the tropics. The main vector involved in the transmission to humans is *Ae. aegypti*. However, other mosquito species were reported to be competent for transmitting YFV to humans [5]. Symptoms of yellow fever (YF) include fever, headache, chills, myalgia, low back pain, vomiting, nausea, fatigue, multisystem organ failure, hemorrhage and shock with renal failure, and hepatitis with jaundice [50]. YFV is estimated to cause about 200,000 cases of disease and 30,000 deaths yearly, with about 90% occurring in Africa. Approximately 20% to 50% of infections developing a severe disease result in fatal outcomes. Yellow fever outbreaks which often occur when the virus is introduced to densely populated urban areas can have disruptive effects on economies and health care systems [51].

It is the first arbovirus investigated and detected in Ghana. Additionally, it is the first recorded arbovirus outbreak in the country. In 1955 (August and September), the first outbreak of YF in Ghana was reported in the Kintampo district (Brong Ahafo Region). This was a small outbreak leading to three confirmed cases via the histology of liver sections. The virus was then isolated from a non-fatal case. Evidence of seroconversion was provided during this outbreak from a village population. It is important to note that the virus was not isolated from mosquitoes. However, the entomological results suggested that *Aedes africanus* may have been the vector [21]. A vaccination campaign was encouraged alongside strengthening vector control activities targeting mosquito larvae to prevent future outbreaks.

Recently, as of 10 April 2022, a total of 166 probable YF cases (IgM positive), including 71 confirmed, were reported from 13 regions in Ghana [29]. From October to November, 2021, a total of 202 suspected cases of YF, including 70 confirmed cases and 35 deaths, have been reported in four regions in Ghana (Savannah, Upper West, Bono and Oti regions) [28]. Thus, a Case Fatality Ratio (CFR) of 17% was recorded. Out of the 202 suspected cases (age range: 4 months–70 years), 52% were females, and 48% were males. The regional laboratory in Senegal (Institute Pasteur Dakar) processed the first three positive PCR test results, which informed the beginning of the outbreak. Furthermore, a total of 70 positive results out of 196 tests were reported using PCR and/or IgM detection. Moreover, plaque reduction neutralization testing was also positive in five samples at the regional reference laboratory [28].

#### 4.1.2. Dengue Virus

This virus is also a member of the *Flaviviridae* family, and *Ae. aegypti* is the main vector [5]. Symptoms of dengue usually last for 2–7 days and include fever, vomiting, nausea, rash, muscle and joint pain, aches and pain behind the eyes. Severe manifestations occur 24–48 h after the fever decreases including fatigue, irritability, tenderness to touch, repeated vomiting, abdominal pain, clots in stool, vomiting of blood, and bleeding from the nose or gums [50].

Currently, there is no evidence indicating outbreaks of dengue in Ghana. However, there is a high risk due to the closeness and high density of the *Aedes* mosquito population in the country. In 2019, the last serological detection of DENV was reported in a study involving adults attending a secondary hospital in the Central Region [39]. The first suspected transmission of DENV via serology was reported after the isolation of DENV-2 from Finnish travelers who visited Ghana in 2005 [30]. Although serological investigations are not a confirmation for virus detection, the positive diagnosis of two children and four adults suggests possible local transmission in Ghana [36,37].

It is important to note that the number of people who become infected with DENV and the risk of exposure is not clear, especially since the symptoms of dengue may be confused with malaria. Factors such as rapid globalization, the presence of *Aedes* mosquito vectors, case reports from travelers, and seroprevalence surveys point toward West Africa as a developing front for dengue surveillance and control [18]. Febrile illnesses are vastly misdiagnosed as malaria in many African settings. Therefore, efficient health care utilization depends on the proper diagnosis of febrile illnesses in the region [18].

### 4.2. Entomological Investigations

Mosquitoes, ticks, sand flies, and biting midges are known to harbor many pathogens and are also involved in their transmission.

Mosquitoes were the most researched vectors with the aim of detecting arboviruses of public health importance. Boorman and colleagues performed the first entomological assessment of the risk of transmission of mosquito-borne viruses in the Ashanti and Brong Ahafo regions in Ghana [21]. During their study, although YFV was not isolated, the results suggested that *Ae. africanus* may have been the vector [21]. Later, other entomological investigations were done to assess the status and risk of transmission of mosquito-borne viruses [44,45,46]. Overall, no arbovirus was detected in any of these entomological studies. However, a possible outbreak of arboviruses was suggested due to the presence of *Aedes* mosquitoes [52]. The detection of arboviruses requires high numbers of mosquitoes for analysis, especially when infection rates are low, and samples have to be stored at −20/−80 °C for the preservation of RNA. This makes arbovirus surveillance expensive and sometimes challenging to implement.

Ticks are responsible for the transmission of arboviruses, which are known as tick-borne viruses. Crimean-Congo hemorrhagic fever virus (CCHFV) associated with livestock was the first tick-borne virus reported in Ghana [43], which was followed by Dugbe and Odaw viruses [48,49]. All these viruses were investigated in the Greater Accra region, and detection was confirmed via RT-PCR. Recently, tick-borne pathogens from domestic animals (cattle, goats, and dogs) were also investigated in Ghana [47]. The morphologically identified ticks were analyzed for pathogens such as CCHFV and Alkhurma hemorrhagic fever virus (AHFV). Interestingly, no RNA of CCHFV or AHFV was detected [47].

Mosquitoes and ticks are prioritized in monitoring programs for vectors, neglecting other blood-feeding vectors such as sand flies and biting midges. Protozoan parasites that cause leishmaniasis and phleboviruses are transmitted by sand flies [53,54]. Examples of sand fly-borne phleboviruses relevant to public health include Ntepes, Naples, Sicilian, and Toscana viruses. Sand fly-borne phleboviruses may cause a transient febrile illness (sand fly fever) or more severe neuroinvasive disease. For example, human meningitis and encephalitis are known to be caused by the Toscana virus [55,56,57]. Sand fly-borne phleboviruses were mainly reported in the Mediterranean region. A study in Portugal reported the co-circulation of a novel phlebovirus (Alcube virus) and Massilia virus in sand flies [58]. In addition, in East Africa (Kenya), the Ntepe virus, isolated in sand flies and specific neutralizing antibodies were found in human serum samples [53]. The first isolated arboviruses from Phlebotomine sand flies in West Africa are Chandipura virus (a vesiculovirus) and Saboya virus (a flavivirus) [59]. Chandipura virus was later confirmed in Asia (India) to be transmitted by sand flies (*Sergentomyia* spp.) [60]. There is, therefore, a scarcity of data in the sub-Sahara African region regarding phlebovirus ecology. In Ghana, entomological surveys revealed the presence of sand flies, especially in leishmania endemic areas [61]. However, there is no available information on sand fly-associated viruses or biting midges-associated viruses in the country. This is a research gap that must be addressed.

### 4.3. Causes of Low Report of Arbovirus Infections in Ghana 

Several factors which can be linked with the environment, host, vector, population, or climate could be responsible for the low numbers of arboviral infections reported in Ghana. However, the list of factors described in this review is not exhaustive and should be considered in any relevant forum on arboviruses and public health. 

#### 4.3.1. Vector Competence and Vectorial Capacity

The ability of a vector to acquire, maintain, and transmit an arbovirus is termed vector competence [8]. Whilst vectorial capacity is a measure of the transmission potential of a vector [62]. Both vector competence and vectorial capacity are often used interchangeably to describe how a vector, for example, a mosquito, could serve as a disease vector. However, vectorial capacity is quantitatively defined. The density of the vector, the vector longevity, and the vector competence influence the vectorial capacity [63]. There is a high possibility that vector competence and vectorial capacity are predisposing factors to the low number of reported cases or prevalence of arboviruses in Ghana. The transmission potential of Ghanaian mosquito population is probably weak. The weaker the transmission potential of a vector, the lower the prevalence of transmitted arbovirus by the respective vector. The competence of *Ae. albopictus* mosquitoes for YFV is low compared with DENV, which could be why YFV never invaded Asia [64]. One major limitation is that experimental vector competence studies involving arboviruses are lacking in Ghana. To the best of our knowledge, only one report is available, which was completed in Japan using Ghanaian mosquitoes [65]. This implies that the lack of expertise and infrastructure are significant limitations. Therefore, qualified staff and high-level biosecurity laboratories for infection experiments are necessary. 

#### 4.3.2. Misdiagnosis of Febrile Illnesses

Most acute febrile patients are often misdiagnosed with malaria due to similar symptoms, such as fever manifested in cases of malaria and certain arboviral infections [66]. It may also be due to inadequate testing capacity for febrile illnesses. It was proposed that certain approaches to acute febrile illness etiology, diagnostics, and management could lead to an increase in health improvements in Africa [67]. This improvement will lead to adequate preparedness for future epidemics of emerging and re-emerging infections such as Ebola, dengue, chikungunya, yellow fever, and probably Zika [67]. A couple of studies highlighted misdiagnosis and co-morbidity with other diseases. It is known that arboviral infections and malaria are both vector-borne diseases, and information about their incidence rates and frequency of co-infection is scarce, together with overlapping geographic distribution [68]. A study in Nigeria, West Africa, reported that arboviral infections have similar symptoms as malaria [66]. This is also confirmed by a study in Senegal, where concurrent malaria and arbovirus infections were reported [68]. A similar study in Ghana revealed that although malaria parasites were detected in febrile pediatric inpatients, flaviviruses and alphaviruses were absent [69]. This may be due to the fact that these arboviruses were simply not present or less-sensitive methods were used. Standardized, certificated, and well-established methods are needed. It is therefore important to properly diagnose infectious pathogens, especially those with the same clinical symptoms such as fever, in order to be able to treat and control them.

#### 4.3.3. Presence of Microbiota in the Mosquito Vector

The mosquito is known to contain a group of organisms such as mosquito-specific viruses (MSVs), *Wolbachia*, bacteria, and fungi. These groups of organisms are referred to as the microbiome. Some of these organisms, such as MSVs and *Wolbachia,* are known to inhibit the replication of some arboviruses. Hence, they are considered a potential biological control tool against major mosquito-borne infections [8]. It has been shown that MSVs may affect the competence of the mosquitoes to transmit these viruses [8]. Therefore, the authors of this review speculate that the microbiota of mosquitoes in Ghana could be responsible for hindering the replication of arboviruses and that the respective mosquito vectors are simply not competent enough to transmit the target arboviruses. Hence, the absence of outbreaks of major arboviruses of public health importance, such as DENV in the country. We propose that more studies based on the interference of MSVs with medically important arboviruses using mosquitoes reared in Ghana are needed. Recently, MSVs were isolated from mosquitoes in Ghana through an entomological investigation [45]. These MSVs should be thoroughly investigated regarding their interaction with arboviruses.

A few studies reported the interference of endosymbiont bacteria *Wolbachia* with arboviruses by decreasing host cytoskeletal proteins and lipids essential for arboviral infection. *Wolbachia* was found to increase host immunity, cellular regeneration and causes the expression of microRNAs, which could potentially be involved in virus inhibition [70]. This implies that *Wolbachia* has antiviral properties via its pathogen-blocking effect to reduce the competence of mosquitoes for arbovirus transmission [70,71]. Currently, it is confirmed that mosquitoes can be infected in nature by *Wolbachia*. It is possible that mosquitoes in Ghana are hosting *Wolbachia* as part of their microbiome, hence, affecting the transmission of arboviruses. Thus, the probable *Wolbachia*-infected mosquitoes in Ghana could be utilized for control programs. 

#### 4.3.4. Mosquito Immune Response 

The replication of arboviruses in their arthropod vectors is controlled by innate immune responses [72]. A variety of different responses such as nodulation, phagocytosis, encapsulation, and signaling pathways (Janus kinase-signal transducer and activator of transcription (JAK-STAT) and the Toll and immune deficiency (Imd)) comprise the innate immune system of mosquitoes [73]. Nevertheless, RNA interference (RNAi) is believed to play the main role in antiviral defense.

The balance between arbovirus and mosquito can be destroyed by a reduced immune system leading to the pathogenicity of the arbovirus instead of the typical persistent infection, which is believed to be vital for the transmission of arboviruses. Inversely, an increased immune system would enable the mosquito to target and clear the arbovirus successfully [8].

RNA interference (RNAi) is a biological process in which RNA molecules are involved in the sequence-specific suppression of gene expression by double-stranded RNA through translational or transcriptional repression [74,75]. RNAi is an important process used by many different organisms, such as mosquitoes, to regulate the activity of genes. The geographical environment of mosquitoes could affect RNAi. Therefore, the diverse immune response could affect the competence of mosquitoes for arbovirus infections. RNAi machinery factors (proteins) such as dicer-2 (dcr-2) and argonaute-2 (ago-2) are usually investigated to determine the effect of the RNAi pathway [76].

DENV replication is modulated by RNAi at different infection stages because the knockdown of key RNAi factors dcr-2 and ago-2 in *Ae. aegypti* mosquitoes affects the prevalence and dissemination of the virus from the midgut to the salivary glands. Virus titer and transmission via saliva are also affected [77]. Other investigations reported that the knockdown of ago-2 limits the replication of Chikungunya virus [72] and Semliki Forest virus [78] in *Ae. aegypti*-derived Aag2 cell lines. Mosquitoes are genetically and geographically diverse, and it is possible that the immune response of mosquitoes in Ghana could greatly interfere with arbovirus infections. This is because there is a strong link between environmental factors and the mosquito immune system [79]. In addition, there is a variation in immune responses in wild mosquito populations [79]. Taken together, the expression of the RNAi factors could be affected due to the complex biotic and abiotic factors influencing the immune system of the mosquito.

#### 4.3.5. Use of Vector Control Tools

Vector control is the main strategy for solving many of the world’s major infectious diseases. Lives have been saved, and the health of millions has been protected when effective methods of targeting mosquitoes, flies, ticks, bugs, and other vectors that transmit pathogens are well implemented [80]. Mitigation strategies against mosquito-borne viruses rely on vector control, although vaccine development for the prevention of these viruses has received great attention. Indeed, vector control could not prevent recent epidemics of major arboviruses such as dengue in some countries. However, it is vital to optimize innovative strategies to control mosquito-borne arboviruses [81]. The use of synthetic chemicals with quick action of killing adult vectors is the primary strategy for outbreak control such as dengue. Some insecticide-treated materials (ITMs) can also protect humans by killing or repelling the vectors [81].

In 1992, a consensus by the World Health Organization (WHO) mentioned insecticide-treated nets (ITNs) as the most promising preventive measure against infections such as malaria. Thereafter, Binka and colleagues in Ghana performed a randomized controlled trial between July 1993 and June 1995 to understand the impact of permethrin-impregnated bednets on child mortality [82]. The trial reported that the use of permethrin-impregnated bednets was associated with a 17% reduction in all-cause mortality in children [82]. Since 2000, mass and continuous distribution channels in Ghana have significantly increased ITN access [83,84]. It is possible that ITNs could prevent the bites of mosquitoes and hence reduce the incidence of arbovirus infections. However, this assumption is not yet confirmed, as no trial was conducted in relation to arboviruses.

In the late 1990s, there was a reduction in YF outbreaks in the Ashanti Region, Ghana. The possible contributing factors could be the presence of other predators consuming *Ae. aegypti* mosquito larvae (*Toxorhynchites brevipalpis* preference for *Ae. aegypti* larvae), the low larval indices and the low host-vector contact rates, and high prevalence of YF antibodies found in the blood of the host population. At that time, the *Tx. brevipalpis* mosquito was found exclusively in the Ashanti region [85].

The new WHO Global Vector Control Response (GVCR) for 2017–2030 outlined a strategic approach to reduce the burden and threat of vector-borne diseases through effective, locally adapted, and sustainable vector control. This approach will not attack a single disease but will target multiple vectors and diseases. This approach will use resources cost-effectively to yield sustainable results [80]. It is also good for a country such as Ghana to deploy this approach appropriately to mitigate the impact of arboviruses and other diseases such as malaria, onchocerciasis, and leishmaniasis.

Information on Integrated Vector Control (IVC) and Integrated Vector Management (IVM) programs, such as long-lasting insecticide-treated nets (LLINs), residual spraying, and larval control, are not readily available in the country. IVC is a strategy to prevent vector-borne diseases by directly targeting the vector that transmits the disease. This involves a decision-making approach to optimally use resources for vector control, and it is based on the principle that to effectively control vectors and the disease they transmit, it requires the partnership and engagement of communities and other stakeholders. Advocacy, social mobilization, legislation, and capacity building are also factors to consider for the adequate implementation of IVM [86,87]. Although a low incidence of arboviruses is reported in Ghana, it is not known what will emerge in the near future. In addition, other vector-borne diseases, such as malaria, are still endemic and of great concern. Hence, IVM is key to eradicating vector-borne diseases.

#### 4.3.6. Ecology and Climate 

Human and environmental drivers affect the dynamics of vector-borne diseases, especially mosquito-borne infections. A major factor for arbovirus exposure is the ecology of the mosquito vector. Humans have altered ecosystems worldwide, and this change impacts infectious diseases such as mosquito-associated pathogens’ transmission in humans and animals [5,88,89]. Ghana has six main agroecological zones (rain forest, transitional zone, Sudan savanna, deciduous forest, Guinea savanna, and coastal savanna) and shares borders with Côte d’Ivoire to the west, Burkina Faso to the north, Togo to the east and the Atlantic Ocean to the south [90]. At least one agroecological region is shared with a neighboring country. However, Ghana has specific agroecological zones. For example, the tropical rainforest zone is shared with Côte d’Ivoire, whilst the Sudan savanna is shared with Burkina Faso. Meanwhile, large local outbreaks of dengue fever were recorded in Côte d’Ivoire and Burkina Faso [91]. Recently, Burkina Faso recorded a large dengue fever outbreak in West Africa in 2016–2017 [92].

Changes in land use are often associated with the emergence of endemic pathogens, as they modulate the interactions and abundance of wildlife and domestic hosts, vectors, and humans [93]. In Ghana, Afrane and colleagues suggested that urban agriculture practice in inland valleys might naturally yield more mosquitoes [94]. Additionally, maize plantations might also influence larval development, the process of pupation, and adult size [95].

Globally, climate change is a growing concern, and there is a high possibility that it will change the burden and distribution of vector-borne infections. Therefore, this poses a threat to public health. Mordecai and colleagues, in their review, projected that climate change could swing the disease burden from malaria infection to arboviruses in Africa [3]. They also postulated that a warming climate will become less appropriate for malaria but more suitable for arboviruses. Hence, a hotspot for arboviruses such as dengue and chikungunya is predicted to increase from the western to the sub-Saharan regions of Africa [3]. Malaria is a long-standing public health threat in Ghana. If these predictions and assumptions become a reality, then Ghana may soon have a share of dengue outbreaks. Although the situation is currently mysterious and unclear, the mosquito vector ecology and climate change may be factors causing the low incidence of arbovirus infections in the country. 

#### 4.3.7. Genetic Diversity of Mosquito Vectors

Mosquitoes have one of the highest levels of genetic diversity amongst eukaryotic organisms. Understanding the genetic diversity of mosquitoes will reveal the role of the vector populations in spatial patterns of arbovirus transmission and distribution [96,97]. Arbovirus-vector–host interactions, markers associated with novel virus emergence and the prediction of arbovirus outbreaks could be revealed by vector genetic diversity investigations. Such studies could also expose the adaptations of arboviruses to different species of hosts and vectors, virulence markers, virus inhibitors resistance, and the effect on the immune system of the host or vector. Several reports used microsatellites, single nucleotide polymorphism (SNP), the NADH dehydrogenase subunit 4 (ND4) gene, and Cytochrome C Oxidase 1 (CO1) gene to genotype various mosquito species [98,99]. A study in Senegal widens our understanding of the global phylogeny of *Ae. aegypti*, indicating that *Aedes aegypti aegypti* (Aaa) and *Aedes aegypti formosus* (Aaf) from West Africa are monophyletic and that Aaa evolved in West Africa from an Aaf ancestor [99]. Interestingly, the study by McBride and colleagues showed that the mosquito’s preference for humans is different according to the mosquito population [100]. Population genetics of mosquitoes in Ghana may contribute to the low report of arboviruses. However, there is a high paucity of data regarding the population genetic diversity of local arthropod vectors such as mosquitoes in Ghana. 

### 4.4. Other Causes of Low Report of Arbovirus Infections in Africa 

The causes of low reported cases of arboviruses in Ghana discussed could be generalized to Africa. Braack and colleagues in their review mentioned some predisposing factors favoring the survival, spread, and prevalence of mosquito-borne viruses [7]. However, other causes but not limited, are ascribed to the whole of Africa. These are protective effects of herd immunity or cross-reactive antibodies; the difference in pathogenicity of strains of viruses occurring in Africa/Ghana compared to strains occurring outside of Africa; vector mutations and adaptations; and virus mutations.

## 5. Future Prospects and Conclusions

The prospects herein discussed highlight gaps identified mainly regarding the causes of low numbers of reported cases of arbovirus infection in Ghana. 

Vector competence studies, for example, experimental infection studies with arboviruses of public health concern are lacking. To the best of our knowledge, only one study was performed with Ghanaian mosquitoes. The study was aimed at determining the vector competence of *Ae. aegypti* for transmitting DENV-1 and DENV-2. The study revealed that all examined Ghanaian mosquitoes were refractory to infection by DENV-2, while some colonies exhibited the potential to transmit DENV-1 [65]. Ghana needs a high-level biosecurity containment facility (Biosafety Level 3, BSL 3) specifically for arbovirus infection experiments. In Ghana, a number of entomological reports involving mosquitoes are available. The abundance and distribution of *Aedes* mosquitoes, which is the main vector for arboviruses of public health threat, was reported. However, vector competence studies for this vector with arboviruses are lacking. To provide an early warning signal, studies of this nature are warranted for emergency preparedness.

Studies involving sand fly-borne viruses and pathogens in biting midges are lacking. This is an open gap that should be thoroughly investigated. Biting midges were not surveyed and/or reported in Ghana to the best of our knowledge. *Culicoides* biting midges are known to harbor arboviruses such as Bluetongue virus (BTV), Schmallenberg virus (SBV), and Oropouche virus (OROV). To date, OROV is the only arbovirus identified as being primarily transmitted by *Culicoides* to and between humans [101]. Therefore, continuous entomological surveillance to detect arboviruses in sand flies, ticks, biting midges, and other vectors is needed.

To prevent the misdiagnosis of febrile illnesses, the authors propose that arboviruses of public health concern such as DENV, WNV, and CHIKV should be part of the routine laboratory investigation for acute febrile diseases such as malaria. Thus, arbovirus and malaria interface investigations are needed. It is possible that the diagnostic methods used in previous studies were not detecting the viruses or they were not sensitive enough. This calls for strengthening the diagnostic methods for arbovirus detection in Ghana.

Studies to understand the impact of mosquito microbiome are needed to ascertain the fact that some of the organisms can be used as biological control of infectious diseases. In this regard, MSVs and *Wolbachia* should be well investigated. Recently, a few MSVs were identified in Ghana. Several reports showed that MSVs can be used as biological control agents, as they were proven to reduce the replication of arboviruses. In vitro/vivo investigations regarding the interaction of MSVs with arboviruses, such as DENV, are needed.

The RNAi machinery of West African mosquitoes should be well understood to validate the complex interaction between arbovirus replication and mosquito innate immune responses. This will help to implement effective control strategies.

Studies should also focus on population genetics using local mosquitoes. There are limited data on the population genetics of arthropod vectors (especially mosquitoes) of arboviruses in West Africa. This will help reveal the genetic dynamics of local mosquitoes or vectors in general to arbovirus infections.

Ghana should mandate an institution with requisite governmental support to assess the risk of infectious diseases affecting human and animal health and strengthen the capacity for their prevention and control. A citizen science approach for disease surveillance practiced in some parts of the Western world can be adopted in Ghana. In this regard, experts are able to develop applications to report emerging or re-emerging viruses in real time. For example, dead birds are monitored for Usutu virus in Germany. Mosquito and tick identification software (apps) are developed to report mosquitoes and ticks identified in real time in the USA. These vectors are investigated by respective experts and scientific reports are sent to the community or various stakeholders.

The authors propose that IVC and IVM programs should be adopted and well-practised in the country. The use of different methods simultaneously (IVC) is now the ideal vector control approach, and this is the foundation of IVM. Unfortunately, due to the lack of funds, resources, expertise, and proper community engagement, these programs are not considered effective in developing countries.

Taken together, apart from the proposed causes of the low incidence of the arbovirus situation in Ghana, the only thing we can say is that it is mysterious and remains an open question. Indeed, there are many factors influencing the arbovirus situation in Ghana, which is probably a mixture of different factors leading to the low prevalence.

## Figures and Tables

**Figure 1 microorganisms-10-01914-f001:**
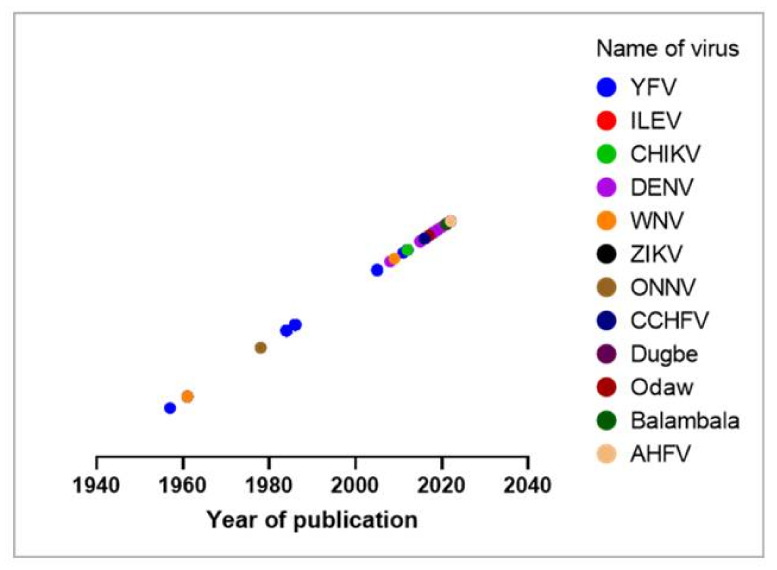
Bubble plot of arboviruses investigated in Ghana (1955–2022). YFV = Yellow fever virus; ILEV = IIesha virus; CHIKV = Chikungunya virus; DENV = Dengue virus; WNV = West Nile virus; ZIKV = Zika virus; ONNV = Onyongnyong virus; CCHFV = Crimean–Congo hemorrhagic fever virus; AHFV = Alkhurma hemorrhagic fever virus.

**Figure 2 microorganisms-10-01914-f002:**
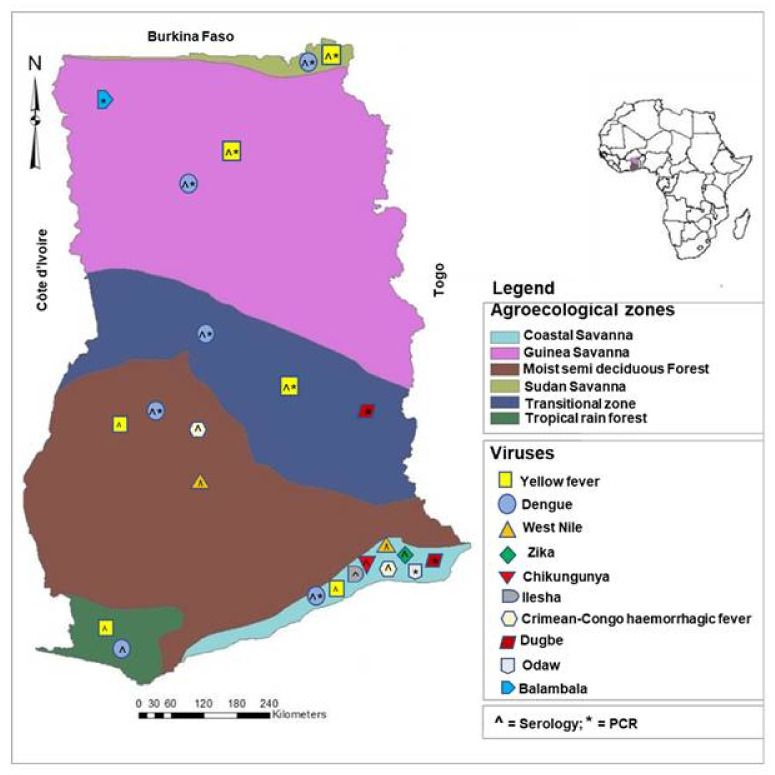
Map of Ghana showing the agroecological zones where the detection of arbovirus antibodies, antigens or RNA was reported. Serology = antibody detection; PCR = antigen or RNA detection. The symbol for serology (^) or PCR (*) is indicated inside the symbol of the relative viruses.

**Table 1 microorganisms-10-01914-t001:** Arboviruses investigated in Ghana (1955–2022).

Year of Sampling/ Detection	Study Design	Number of People/Sample/Cases	Frequencies (Positives/ Deaths)	Region (Place)	Detection Method	Reference
	**Yellow fever (*Flaviviridae*)**			**Human cases**		
1955	Cross-sectional	12 cases, 155 total sera	3 confirmed	Brong Ahafo (Kintampo)	Histology, Serology	[21]
1959	Cross-sectional	76 sera	38.3% CFR	Greater Accra (Tema)	Serology—Complement fixation	[22]
1963	Case report	3 cases	NI	Ashanti (/Kumasi), Northern (Damongo)	NI	[23]
1969	Case report	5 cases 303 cases	3 deaths 72 deaths	Northen (Tamale) Upper East (Bolgatanga)	NI	[23]
1970	Case report Case report	11 cases NI	14 deaths 60 deaths	Eastern (Akwatia) Eastern (Asikasu)	NI Serology, Histopathology	[23] [24]
1970–1975	Case report	12 cases	7 deaths	Brong Ahafo (Dormaa Ahenkro, Berekum, Hwidiem)	NI	[23]
1977–1978	Cross-sectional	136 cases	34 deaths	Upper East (Jirapa)	Serology, Histopathology	[23]
1978–1979	Cross-sectional	239 cases 340 cases	56 deaths 52 deaths	Eastern (Maase, Somanya, Akuse, Akosombo, Nkwakwa, Asamankese) Volta (Hohoe, Kpandu)	Histopathology Serology, Histopathology	[23]
1979–1980	Cross-sectional	104 cases	41 deaths (39% CFR)	Brong Ahafo (Wenchi, Techiman, Hwidiem, Berekum, Dormaa Ahenkro) Eastern (Akwatia)	Serology, Histopathology	[23]
1979	Case report	NI	4 deaths	Greater Accra	Serology, Histopathology	[25]
1980	Case report	6 cases 2 cases	4 deaths 2 deaths	Brong Ahafo Volta	Serology, Histopathology	[24]
	**Yellow fever (*Flaviviridae*)**			**Human cases**		
1981	Case report	6 cases	3 deaths	Whole Ghana	Serology, Histopathology	[24]
1982	Case report	7 cases	5 deaths	Whole Ghana	Serology, Histopathology	[24]
1983	Case report	205 cases 53 cases 55 cases 12 cases	120 deaths 36 deaths 8 deaths 12 deaths	Northern (Bole, Damongo, Gambaga, Tamale) Upper East (Bawku, Bolgatanga) Upper West (Wa, Tumu) Brong Ahafo (Kintampo)	Serology, Histopathology	[24]
1983–1984	Case report	372 cases	201 deaths	Upper regions (Jirapa, Wa, Bolgatanga, Navrongo, Nandom, Jirapa)	Serology, Histopathology	[25]
1993–1994	Case report	118 cases	26 deaths	Upper West (Jirapa)	Serology, Histopathology	[25]
1996–1997	Case report	33 cases	5 deaths	Upper East (Bolgatanga etc.), Northern (Mamprusi)	Serology, Histopathology	[25]
2005	Case report	3 cases	None	Upper West (Jirapa, Wa, Nadowli)	NI	[26]
2011	Case report	2 cases 3 cases	1 death None	Northern region/Sawla Kalba Greater Accra (Ledzokuku)	NI	[26]
2016	Case report	4 suspected	None	Brong Ahafo, Volta	NI	[27]
2021	Case report	70 cases	35 deaths (17% CFR). PCR positive	Savannah, Upper West, Bono, and Oti	Serology—IgM, PCR, Plaque reduction neutralization assay.	[28]
2022	Case report	71 cases	71 IgM positive	Whole Ghana (13 regions)	Serology	[29]
	**Dengue (*Flaviviridae*)**			**Human cases**		
1959	Cross-sectional	76 sera	Not specific	Greater Accra (Tema)	Serology	[22]
2005	Cross-sectional	11 isolates	1 PCR positive	Ghana-Finland	Serology—IgG/IgM, RT-PCR	[30]
2011–2014	Cross-sectional	218 children	3.2% IgM, 21.6% IgG	Greater Accra (Kpeshie), Brong Ahafo (Kintampo), Upper East (Navrongo)	Serology—IgG/IgM, RT-PCR	[31]
2012–2014	Cross-sectional	236 sera	87.2% antibody prevalence	Greater Accra (Korlebu)	Serology, Plaque reduction neutralization tests (PRNT)	[32]
2013	Cross-sectional	360 sera	1.9% IgM, 3.6% IgG	Whole Ghana (ten regions)	Serology—IgG, IgM, RT-PCR	[33]
2013–2015	Cross-sectional	188 sera	82 (43.6%) IgG	Ashanti (Agogo, Kumasi), Brong Ahafo (Techiman)	Serology—IgG, IgM, RT-PCR	[34]
2014	Cross-sectional	417 sera	29.7% IgG	Whole Ghana (ten regions)	Serology—IgG	[35]
2014–2016	Cross-sectional	150 patients	32 IgM positive, 4 PCR positives	Greater Accra, Central, Upper East	Serology—IgG, IgM, RT-PCR	[36]
2016–2017	Cross-sectional	700 children	2 PCR positives, IgG/IgM positive	Greater Accra, Brong Ahafo	Serology—IgG, IgM, RT-PCR	[37]
2016–2017	Cross-sectional	260 febrile patients	69.23% antibody positive	Greater Accra	Serology—IgG, IgM, NS1, RT-PCR	[38]
2019	Cross-sectional	270 participants	12.6% IgG, 2.2% IgM positives	Central (Cape Coast, Komenda)	Serology—IgG/IgM	[39]
**Human cases**						
**West Nile (*Flaviviridae*)**						
1959	Cross-sectional	76 sera	Not specific	Greater Accra (Tema)	Serology—Complement fixation	[22]
2009 *	Cross-sectional	1324 plasma	Children = 4.8% IgG, 2.4% IgM positive	Ashanti (Kumasi)	Serology—IgG/IgM, PCR	[40]
**Zika (*Flaviviridae*)**						
2012–2014	Cross-sectional	236 sera	12.9% antibody positive	Greater Accra (Korlebu)	Serology—IgG/IgM, PRNT	[32]
2016–2017	Cross-sectional	160 patients	20.6% antibody positive	Greater Accra	Serology—IgG/IgM	[41]
**Chikungunya (*Togaviridae*)**						
1959	Cross-sectional	76 sera	Not specific	Greater Accra (Tema)	Serology—Complement fixation	[22]
2016–2017	Cross-sectional	260 patients	27.69% antibody positive	Greater Accra	Serology-NS1/IgG/IgM, RT-PCR	[38]
**Onyongnyong (*Togaviridae*)**						
1954	Cross-sectional	86 travelers to Britain	3 seropositive	NI	Serology—Complement fixation, Hemagglutination inhibition, Immunofluorescence assay	[42]
**Ilesha (*Peribunyaviridae*)**						
1959	Cross-sectional	76 sera	28.5% (30–44 years) positive	Greater Accra (Tema)	Serology—Complement fixation	[22]
**Crimean–Congo hemorrhagic fever (*Nairoviridae*)**						
2011	Longitudinal study	188 sera	5.7% seroprevalence	Greater Accra, Ashanti (Kumasi)	Serology—IgG/IgM	[43]
			**Yellow fever (*Flaviviridae*)**	**Mosquito surveillance**		
1955	Cross-sectional	299 mosquitoes	No arbovirus detected	Brong Ahafo (Kintampo)	Serology, Histology	[21]
1999–2000	Cross-sectional	2804 households	No arbovirus detected	Northern (Damongo), Upper East (Bolgatanga), Upper West (Jirapa, Tumu)	RT-PCR	[44]
			**Dengue (*Flaviviridae*)**			
1999–2000	Cross-sectional	2804 households	No arbovirus detected	Northern (Damongo), Upper East (Bolgatanga), Upper West (Jirapa, Tumu)	RT-PCR	[44]
2015–2016	Cross-sectional	36 mosquitoes per pool	Only mosquito-specific virus detected	Greater Accra. Volta. Western. Ashanti. Upper West. Savannah.	RT-PCR	[45]
2018–2019	Cross-sectional	1930 *Aedes* mosquitoes, 75 pools.	No arbovirus detected	Northern (Larabanga, Mole)	RT-PCR	[46]
			**Zika (*Flaviviridae*)**			
2018–2019	Cross-sectional	1930 *Aedes* mosquitoes, 75 pools.	No arbovirus detected	Northern (Larabanga, Mole)	RT-PCR	[46]
			**West Nile (*Flaviviridae*)**			
2015–2016	Cross-sectional	36 mosquitoes per pool	Only mosquito-specific virus detected	Greater Accra. Volta. Western. Ashanti. Upper West. Savannah.	RT-PCR	[45]
			**Chikungunya (*Togaviridae*)**			
2015–2016	Cross-sectional	36 mosquitoes per pool	Only mosquito-specific virus detected	Greater Accra. Volta. Western. Ashanti. Upper West. Savannah.	RT-PCR	[45]
2018–2019	Cross-sectional	1930 *Aedes* mosquitoes, 75 pools.	No arbovirus detected	Northern (Larabanga, Mole)	RT-PCR	[46]
**Tick surveillance**						
**Crimean-Congo haemorrhagic fever (*Nairoviridae)***						
2011	Longitudinal	144 ticks, 97 pools	5 positive pools	Greater Accra, Ashanti (Kumasi)	Serology—IgG/IgM, RT-PCR	[43]
2016–2017	Cross-sectional (domestic dogs, goats, and cattle)	2016 ticks, 912 pools	No CCHFV detected	Greater Accra (Accra), Northern (Tamale)	RT-PCR	[47]
**Dugbe (*Nairoviridae)***						
2015	Cross-sectional (domestic dogs and cattle)	153 ticks, 29 pools	2 positive pools	Greater Accra (Mobore)	RT-PCR	[48]
2016	Cross-sectional (domestic dogs and cattle)	354 ticks, 93 pools	1 positive pool	Volta (Hohoe)	RT-PCR	[49]
**Odaw (*Phenuiviridae)***						
2015	Cross-sectional (domestic dogs and cattle)	153 ticks, 29 pools	4 positive pools	Greater Accra (Pokuase, Korle-Gonno)	RT-PCR	[48]
2016	Cross-sectional (domestic dogs and cattle)	354 ticks, 93 pools	4 positive pools	Greater Accra (Accra)	RT-PCR	[49]
**Balambala (*Phenuiviridae)***						
2016	Cross-sectional (domestic dogs and cattle)	354 ticks, 93 pools	2 positive pools	Upper west (Jirapa)	RT-PCR	[49]
**Alkhurma haemorrhagic fever (*Flaviviridae*)**						
2016–2017	Cross-sectional (domestic dogs, goats, and cattle)	2016 ticks, 912 pools	No AHFV detected	Greater Accra (Accra), Northern (Tamale)	RT-PCR	[47]
**Sand fly/Biting midges surveillance**						
No available record						

* Year of publication of research paper if year of sampling/detection is not specified in the literature; NI = Not indicated in the literature; CFR = Case Fatality Rate.

## Data Availability

Data supporting this review are already enclosed.

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
