# Peer review of "Arbovirus Epidemiology: The Mystery of Unnoticed Epidemics in Ghana, West Africa"

_microorganisms, 2022, doi:10.3390/microorganisms10101914_

Round 1
Reviewer 1 Report
The authors appear to be using the words virus and viral interchangeably. Virus is the noun and viral the adjective.
Line 37 Not sure what is meant by façade. Surely there is a better term.
Line 320
This could be expanded to indicate that as this has been used elsewhere to displace the mosquitoes that are readily transmitting the viruses such as Dengue it may be possible to find some in Ghana that are infected and utilize them in a control program.
Line 380
I assume this is Flavivirus antibodies.
The table is quite large and maybe this could be included as supplementary data.

Reviewer 2 Report
I'm convinced that this work is meaningful, well done and scientifically and certainly correct.
In particular the discussion is innovative and very complete.
The citizen science approach used in this Paper and the goodness and richness of the work done therefore imposes in point 5 for the conclusions at least one sentence with an even stronger impact due to little sense of responsibility that I feel for a more significant use of adequate resources by the competent Authorities.
